

# Comparison of prediction power of three multivariate calibrations for estimation of leaf anthocyanin content with visible spectroscopy in *Prunus cerasifera*

Xiuying Liu[1,2,3], Chenzhou Liu[1,2], Zhaoyong Shi[1,2] and Qingrui Chang[4]

[1] College of Agriculture, Henan University of Science and Technology, Luoyang, Henan, China
[2] Luoyang Key Laboratory of Symbiotic Microorganism and Green Development/Luoyang Key Laboratory of Plant Nutrition and Environmental Ecology, Luoyang, Henan Province, China
[3] Research Center of Forestry Remote Sensing and Information Engineering, Central South University of Forestry and Technology, Changsha, Hunan Province, China
[4] College of Resources and Environment, Northwest A&F University, Yangling, Shaanxi Province, China

Corresponding authors
Xiuying Liu, liuxy@haust.edu.cn, csfulxy@126.com
Qingrui Chang, chqrui@126.com

## ABSTRACT

The anthocyanin content in leaves can reveal valuable information about a plant's physiological status and its responses to stress. Therefore, it is of great value to accurately and efficiently determine anthocyanin content in leaves. The selection of calibration method is a major factor which can influence the accuracy of measurement with visible and near infrared (NIR) spectroscopy. Three multivariate calibrations including principal component regression (PCR), partial least squares regression (PLSR), and back-propagation neural network (BPNN) were adopted for the development of determination models of leaf anthocyanin content using reflectance spectra data (450–600 nm) in *Prunus cerasifera* and then the performance of these models was compared for three multivariate calibrations. Certain principal components (PCs) and latent variables (LVs) were used as input for the back-propagation neural network (BPNN) model. The results showed that the best PCR and PLSR models were obtained by standard normal variate (SNV), and BPNN models outperformed both the PCR and PLSR models. The coefficient of determination ($R^2$), the root mean square error of prediction ($RMSE_p$), and the residual prediction deviation (RPD) values for the validation set were 0.920, 0.274, and 3.439, respectively, for the BPNN-PCs model, and 0.922, 0.270, and 3.489, respectively, for the BPNN-LVs model. Visible spectroscopy combined with BPNN was successfully applied to determine leaf anthocyanin content in *P. cerasifera* and the performance of the BPNN-LVs model was the best. The use of the BPNN-LVs model and visible spectroscopy showed significant potential for the nondestructive determination of leaf anthocyanin content in plants.

# INTRODUCTION

Anthocyanins are a large group of water soluble flavonoid pigments (*Strack, 1997*; *Iwashina, 2000*), the common pigment, that occur in all tissues of higher plants, including the leaves,

stems, roots, flowers, and fruits. They are responsible for a wide range of plant colors, such as blue, purple, violet, magenta, red and orange (*Fennema, 1996*; *Lai et al., 2019*), but they often appear red (*Gould et al., 1995*; *Van Den Berg & Perkins, 2005*; *Gould, Davies & Winefield, 2009*). Anthocyanins serve many functions, including pollinator attraction, as protectants (*Gould, Davies & Winefield, 2009*), as antioxidants (*Gould, McKelvie & Markham, 2002*; *Yang et al., 2017*), and as osmoprotectants (*Chalker-Scott, 1999*). These compounds also play a photo-protective role (*Liakopoulos et al., 2006*), and act as optical barriers (*Close & Beadle, 2003*; *Solovchenko & Merzlyak, 2008*). A number of environmental stresses, such as strong light, low temperature, UV-B irradiation, wounding, drought, bacterial and fungal infections, deficiencies in nitrogen, phosphorus and potassium, and certain herbicides and pollutants can result in the significant accumulation of anthocyanins (*Saure, 1990*; *Garriga et al., 2014*; *Zhang et al., 2018*), which are thus often referred to as "stress pigments" (*Chalker-Scott, 1999*). In addition, anthocyanins accumulate transiently in juvenile and senescing leaves in many plant species under unfavorable conditions (*Karageorgou & Manetas, 2006*; *Merzlyak et al., 2008*; *Zeliou, Manetas & Petropoulou, 2009*; *Garriga et al., 2014*). Thus, anthocyanin content can serve as an indicator of leaf senescence and environmental stresses in many plant species (*Neill & Gould, 1999*; *Gitelson & Merzlyak, 2004*), so the accurate detection and quantitative assessment of anthocyanin can provide important and valuable information about the physiological responses and adaptation of plants to environmental stresses (*Gamon & Surfus, 1999*; *Gitelson, Chivkunova & Merzlyak, 2009*; *Ustin et al., 2009*). The traditional method to determine anthocyanin content has been the wet-chemical method (*Gitelson & Merzlyak, 2004*; *Gitelson, Merzlyak & Chivkunova, 2001*; *Steele et al., 2009*). This method is laborious, time-consuming, expensive, and requires the destruction of leaves for measurement (*Solovchenko et al., 2001*; *Merzlyak, Solovchenko & Gitelson, 2003*; *Steele et al., 2009*). In addition, this measurement method does not allow the measurement of changes in pigments over time in a single leaf (*Garriga et al., 2014*).

Visible and near infrared reflectance (Vis/NIR) spectroscopy has been widely used in recent decades to measure pigments. The spectral absorbance properties of pigments are present in the reflectance spectra of leaves, thus measurements of reflected radiation can be used as a non-destructive method to quantify pigments (*Blackburn, 2007*). Non-destructive technology based on spectrum analysis has several advantages over conventional methods, including simplicity, sensitivity, inexpensive, good reliability of the method, and high performance (*Viscarra Rossel, McGlynn & McBratney, 2006*; *Kira, Linker & Gitelson, 2015*; *Nagy, Riczu & Tamás, 2016*). This technique can be applied at different spatial scales and in a large number of samples (*Viña & Gitelson, 2005*; *Lobos et al., 2014*). Compared with traditional multispectral techniques, hyperspectral remote sensing, which provides a continuous reflectance spectrum with narrow wavebands, can characterize vegetation and provide a considerably greater amount of information than what can be obtained using traditional multispectral techniques (*Goetz, 2009*; *Mulla, 2013*). Therefore, recent research has focused on developing techniques to analyze plant spectra to more accurately quantify pigment concentrations (*Blackburn, 2007*). Most research has focused on the estimation of chlorophyll and carotenoid content, but little is known about anthocyanin estimation
from reflectance spectra, and most pigment measurement studies utilize linear/or simple nonlinear models (*Chappelle, Kim & McMurtrey, 1992*; *Gitelson, Merzlyak & Chivkunova, 2001*; *Blackburn, 2007*). For anthocyanins, various models (called vegetation indices) have been developed based on the spectral information (e.g., *Gitelson, Merzlyak & Chivkunova, 2001*; *Gitelson & Merzlyak, 2004*; *Gitelson et al., 2006*; *Gitelson, Chivkunova & Merzlyak, 2009*; *Van Den Berg & Perkins, 2005*; *Merzlyak et al., 2008*; *Steele et al., 2009*; *Garriga et al., 2014*; *Liu et al., 2015*; *Manjunath, Shibendu & Dhaval, 2016*).

The empirical statistical approach is a main approach to building relationships between spectral data and biochemical or biophysical parameters. The modern spectral technique (especially hyperspectral data) generally produces abundant data for the analyzed object. However multi-collinearity is a common problem inherent to hyperspectral dataset (*Mirzaie et al., 2014*). There are convoluted interrelations between individual values of reflectance and biological properties (*Garriga et al., 2014*). Moreover univariate regression models based on vegetation indices, which typically use two to three bands, cannot capture the intrinsic relationships between the observed remote sensing data (especially hyperspectral data) and biochemical or biophysical parameters of interest (*Camps-Valls et al., 2006*). Furthermore, the selection of calibration method is a main factor influencing measurement accuracy with visible and near infrared reflectance (Vis/NIR) spectroscopy (*Mouazen et al., 2010*). Hence, it is important to use multivariate calibration algorithms to better develop the relationship between spectral data and the analyzed object and compare predictive performance (*Mouazen et al., 2010*; *Li & He, 2010*). Linear and nonlinear multivariate calibration techniques include principal component regression (PCR), partial least squares regression (PLSR), and back-propagation neural network (BPNN), and have been widely and successfully applied in spectra analysis (*Vasques, Grunwald & Sickman, 2008*; *Liu et al., 2008*; *Atzberger et al., 2010*; *Li & He, 2010*; *Kinoshita et al., 2011*; *Mirzaie et al., 2014*; *Gomes et al., 2017*; *Wang et al., 2018*). The PCR and PLSR analyses are the most common techniques for spectral calibration and prediction (*Viscarra Rossel, McGlynn & McBratney, 2006*), and these two methods may reduce the effect of the multi-collinearity problem. The artificial neural network (ANN) has many advantages such as nonlinear mapping, high accuracy for learning, and good robustness (*Atkinson, 1997*; *Keiner & Yan, 1998*). For this reason, artificial neural networks are increasingly used in visible and near infrared reflectance (Vis/NIR) spectroscopy (*Liu et al., 2008*; *Gomes et al., 2017*).

*Prunus cerasifera* (*P. cerasifera*), commonly called cherry plum, is a Prunus deciduous small trees that is natives to western Asia and the Caucasus. Its leaves contain high amounts of anthocyanins, which makes them appear purple. *P. cerasifera* has become a very popular ornamental landscape tree in large part because its showy purple foliage retains excellent color throughout the growing season. The leaves of *P. cerasifera* exhhibit a wide range of anthocyanin contents, making *P. cerasifera* a good object to study the content of leaf anthocyanins in plants. To the best of our knowledge, no work has explored the combination of PLSR or PCR with ANN for the analysis of leaf anthocyanin content of *P. cerasifera* using visible spectroscopy (450–600 nm).

In this study, the leaf anthocyanin content of *P. cerasifera* was investigated with visible spectroscopy based on three multivariate calibrations. The objectives of the present

work were: (1) to investigate the feasibility of using visible spectroscopy to determine the anthocyanin content in *P. cerasifera* leaves; (2) to determine the optimal spectral pretreatments after the comparison of Savitzky-Golay (SG) smoothing, standard normal variate (SNV), multiplicative scattering correction (MSC), first derivative(1-Der), standard normal variate in combination with transformed baseline (SNV+TB), Savitzky-Golay smoothing in combination with first derivative (SG+1-Der), and multiplicative scattering correction in combination with first derivative (MSC+1-Der); (3) to develop the best calibration models to estimate the leaf anthocyanin content in *P. cerasifera* comparing the prediction power of principal component regression (PCR), partial least squares regression (PLSR), and back-propagation neural network (BPNN). The results of this study are a preliminary step forward for improving monitoring of the growing status and biological parameters of plants using spectroscopic techniques.

## MATERIALS AND METHODS

### Leaf samples

In total, 456 pieces of *P. cerasifera* leaves were collected from the Northwest A & F University campus between March and May of 2015. These leaves, ranging in color from dark green with little red to completely red, were picked from *P. cerasifera* of different ages and oriented in different directions from the stem. After detachment, the leaves were immediately sealed in plastic bags with a small amount of water, labeled as different samples, and then placed on ice for transport to the laboratory. Healthy and homogeneously colored leaves without visible symptoms of damage were used for experiments.

### Laboratory analyses of anthocyanin content

The anthocyanin content was quantitatively measured from the same leaf samples used for reflectance measurement. Several small pieces were cut from the leaves and then o.15 g of samples were extracted with 0.1 mol $L^{-1}$ hydrochloric acid methanol solution using the soaking extraction method. For total anthocyanin extraction, 24 h of soaking time was performed. The resulting extracts were immediately assayed spectrophotometrically, and the anthocyanin content was expressed as a function of leaf amount (i.e., $\mu$mol $g^{-1}$). The methods used are described in detail in the literature (*Xiong et al., 2003*).

### Spectrum measurement and pretreatment

The reflectance spectra of the leaves were measured with a SVC HR-1024i spectrophotometer (Spectra Vista Corporation, Poughkeepsie, N, USA) equipped with a SVC reflectance probe and interfaced with a personal computer. During measurement, an internal tungsten halogen lamp provided artificial illumination. The HR-1024i spectrophotometer measures radiance with a spectral resolution of 3.5 nm in a wavelength range of 350 to 1,000 nm. Before measuring the reflectance spectra of the leaves reference measurements were made by rotating the sample holder plate to position the white reference panel facing the probe window. Target measurements were then taken by inserting a leaf between the sample holder plate and the window. For accurate measurement of the reflectance of the leaves, three reflectance measurements were acquired for each leaf and each sample included four

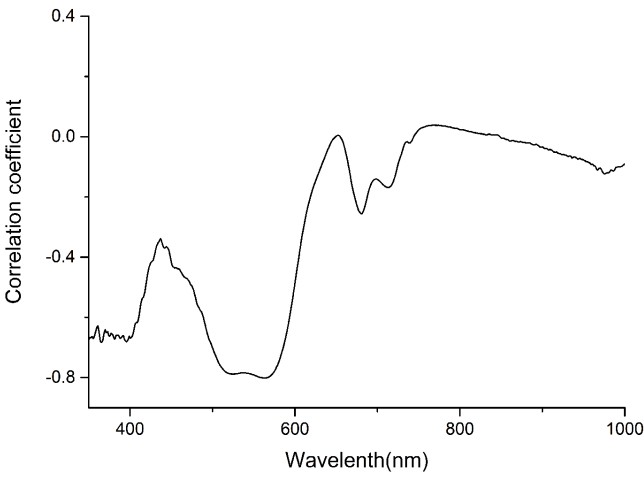

**Figure 1** Correlation coefficient between anthocyanin content and Spectra of *P. cerasifera* leaves.

leaves of the same color. Thus, average values were calculated of twelve spectra sample to establish a single representative reflectance spectrum.

The anthocyanin absorption peaks in situ were around 540–550 nm in the visible/near-infrared (Vir/NIR) rang (*Gitelson, Merzlyak & Chivkunova, 2001*; *Merzlyak et al., 2008*). The analysis showed a high correlation between total anthocyanin content and reflectance spectra between 350 and 600 nm, and relative low correlation at the other wavebands (Fig. 1). Signals in the first 100 nm were removed to avoid a low signal-to-noise ratio. Finally, only wavelength bands between 450 and 600 nm, which avoided the effect of leaf structure and the strongest absorption of chlorophyll and water, were employed for the calculations.

To remove system noises and external disturbances and to select the best pretreatment method, some pretreatments were performed on the spectra and the results were compared (*Liu et al., 2008*; *Liu & Liu, 2013*). The reflectance spectra were first imported into the SVC HR-1024i software (Spectra Vista Corporation, USA). Overlapping detector data were removed, and then resampling in 1 nm intervals was performed. Next, seven types of pretreatments were applied and compared: standard normal variate (SNV), multiplicative scattering correction (MSC), Savitzky-Golay smoothing (SG), first derivative (1-Der), standard normal variate combined with transformed baseline (SNV+TB), multiplicative scattering correction combined with first derivative (MSC+1-Der), and Savitzky-Golay smoothing combined with first derivative (SG+1-Der). SNV, MSC, and SG smoothing were applied to remove the multiplicative effects of scattering, random noise, and spectral baseline shift (*Chu, Yuan & Lu, 2004*; *Zhao, Qu & Cheng, 2004*; *Liu et al., 2008*; *Bao et al., 2012*). The first derivative pretreatment method was applied to decrease the baseline shift (*Liu et al., 2008*). The raw reflectance spectra and preprocessed spectra of *P. cerasifera* leaves are shown in Figs. 2A–2H. All pre-processing steps were implemented using the Unscrambler 9.7 (Camo Inc., Oslo, Norway).
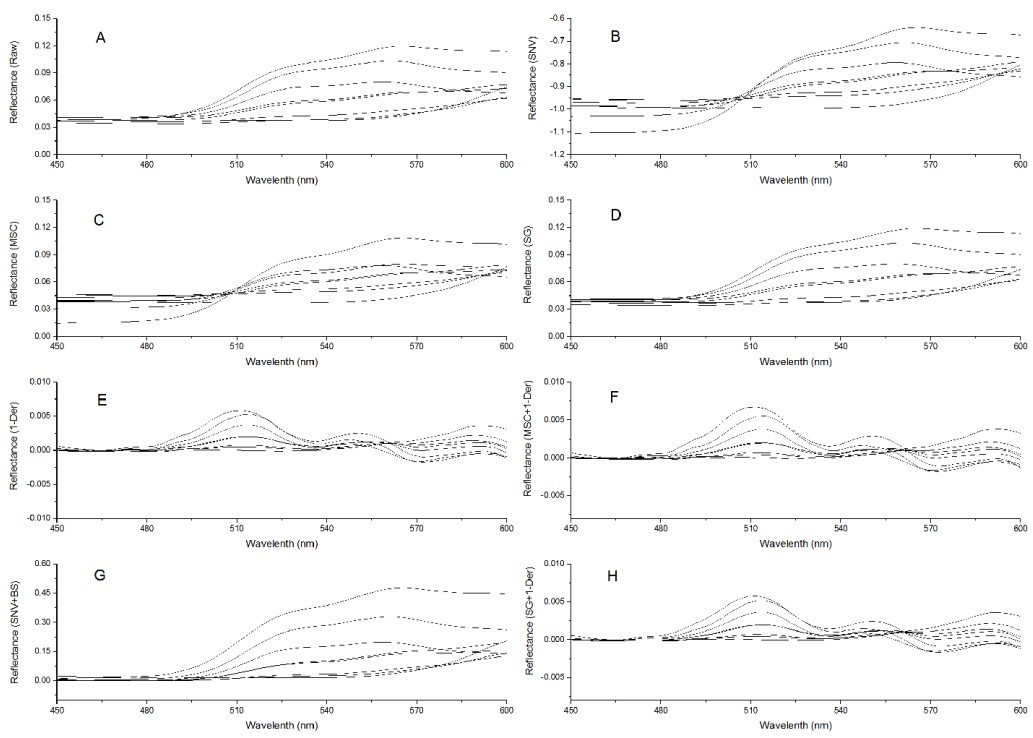

**Figure 2** **Spectra of *P. cerasifera*. leaves.** (A) The raw spectra of *P. cerasifera* leaves; (B) SNV; (C) MSC; (D) SG; (E) 1–Der; (F) MSC+1-Der; (G) SNV+TB; (H) SG+1-Der.

## Establishment of calibration models
### Principal component regression

Principal component regression (PCR) is a method to relate variations in a response variable (Y-variable) to the variations of several predictors (X-variables), with explanatory or predictive purposes. This method performs particularly well when the various X-variables express common information with a high amount of correlation, or even collinearity (*Martens & Naes, 1989*). The optimal number of principal components (PCs) for a model was determined by examining a plot of leave-one-out cross-validation residual variance against the number of loadings (*Mouazen et al., 2010*).

### Partial least squares regression

Partial least squares (PLS) analysis is a bilinear regression method (*Arana, Jaren & Arazuri, 2005*) is widely utilized as a multi-analysis method in spectroscopy (*Soriano et al., 2007*; *Li & He, 2010*; *Wu et al., 2011*; *Zhang et al., 2018*). Partial least-squares regression can reduce data noise and computation time, with only minor loss of the information contained in the original variables. The main procedure is to extract the PLS factors and determine the linear correlationships between the PLS factors and chemical constituents. In the development of the PLS model, leave-one-out cross-validation was used to evaluate the quality and to prevent overfitting of the calibration model (*Mouazen et al., 2010*). All calculations of the PCR and PLSR were also implemented based on the Unscrambler V9.7.

### Back-propagation neural network

The most popular neural network is BPNN, a type of nonlinear neural network used to solve classification and regression problems. BPNN models usually provide better results than traditional statistical methods. However, extreme long training time requirements and over-fitting are two main limitations of ANN calibration when using raw spectral data points or when too many spectral data points are selected as inputs (*Mouazen et al., 2010*). Many studies have shown that adopting PCs or LVs as input for BPNN is an effective way to reduce computation resources and improve the robustness of ANN calibration (*He et al., 2006*; *Janik et al., 2007*; *Mouazen et al., 2010*; *Mirzaie et al., 2014*). Hence, in this study BPNN analyses were performed using LVs obtained from PLSR (BPNN-LVs) and PCs obtained from PCA (BPNN-PCs). The first five PCs (spectra preprocessed by SNV) were considered as input variables in this study, since they could explain nearly 95% of the variance. The first five LVs (spectra preprocessed by SNV) also were applied as input variables of the BPNN model, as the residual variance was the first minimum value (*Brown, Bricklemyer & Miller, 2005*).

A standard three-layer feed-forward network composed of one input layer, one hidden layer, and one output layer (one node) is usually applied for spectral calibration and prediction (*Liu et al., 2008*; *Mouazen et al., 2010*; *Mirzaie et al., 2014*; *Gomes et al., 2017*; *Wang et al., 2018*). Therefore a simple-hidden-layer neural network was used in this study to estimate the anthocyanin content in *P. cerasifera* leaves. Each node in ANN represents a "neuron", and is associated with a transfer (activation) function that sums the outputs from that node and passes them to the next layer in the network. The tan-sigmoid function and a linear function were respectively adopted in the hidden and in the output layers. The numbers of neurons in the hidden layer was optimized by trial-and-error. For network training, we used Levenberg–Marquardt (TRAINLM), and the early stopping technique was used to avoid overfitting problems (*Demuth, Beale & Hagan, 2010*; *Mirzaie et al., 2014*). All BPNN calculations were implemented using the Neural Networks toolbox of MATLAB. The theory of ANN has been described previously (*He et al., 2006*). During training, the number of nodes in the hidden layer was constantly readjusted. When the number of nodes of the hidden layer was set at five, a very good result was achieved. In this way, the BPNN model for anthocyanin content was obtained. The structure contained one input layer with five modes, and the hidden layer contained five nodes and one output node.

To ensure that the calibration or validation set included samples that covered the complete range of each chemical parameter, the 114 sample data (456 pieces of leaves, four leaves per sample) were arranged in ascending order according to anthocyanin content. Arranged from the lowest to the highest value, two of every three samples were selected for inclusion in the calibration set (76) and the remaining one-third of the samples were considered the validation set (38). Therefore, each sample was only used in either the calibration or the validation sets, but not both sets. To compare the performances of different calibration models, the same calibration and validation sets were used to test all of the models. Previous studies have assessed the accuracy and the estimating performance of different models in terms of absolute prediction accuracy (RMSE), the coefficient of determination ($R^2$), and the residual prediction deviation (RPD)

**Table 1 The statistical values of anthocyanin content.**

| Data sets | Sample number | Minimum | Maximum | Mean | Standard deviation |
|---|---|---|---|---|---|
| Calibration | 76 | 0.36 | 4.61 | 1.99 | 0.98 |
| Valibration | 38 | 0.41 | 3.96 | 1.93 | 0.95 |
| All samples | 114 | 0.37 | 4.61 | 1.97 | 0.97 |

(*Saeys, Mouazen & Ramon, 2005*; *Viscarra Rossel, McGlynn & McBratney, 2006*; *Vasques, Grunwald & Sickman, 2008*; *Mouazen et al., 2010*; *Kinoshita et al., 2011*; *Hu, 2013*; *Du et al., 2013*; *Mirzaie et al., 2014*; *Gomes et al., 2017*). In this study, the performance of all models was evaluated by the following indices: the coefficients of determination of calibration ($R^2_{cal}$) and validation ($R^2_{val}$), the root mean square errors of calibration ($RMSE_c$) and validation ($RMSE_p$), and the residual prediction deviations of calibration ($RPD_{cal}$) and validation ($RPD_{val}$). The detailed formulas of these indices are as published previously (*Hu, 2013*). Based on experience and previous reports (*Viscarra Rossel, McGlynn & McBratney, 2006*; *Saeys, Mouazen & Ramon, 2005*), the $R^2$ and RPD values were classified as follows: $R^2 < 0.5$ with $1.0 \leq RPD < 1.4$ indicates poor models/predictions able to distinguish only high and low values; $0.5 \leq R^2 < 0.65$, $1.4 \leq RPD < 1.8$ indicates fair models/predictions which can be used for assessment and correlation; $0.65 \leq R^2 < 0.80$, $1.8 \leq RPD < 2.0$ indicates good models/predictions where quantitative predictions are possible; $0.80 \leq R^2 < 0.90$, $2.0 \leq RPD < 2.5$ indicates very good quantitative models/predictions, and $R^2 \geq 0.90$, $RPD \geq 2.5$ indicates excellent models/predictions. Generally, a good model should have higher $R^2$ and RPD values, and lower RMSE values.

## RESULTS

### Features of spectra

The raw reflectance spectra of *P. cerasifera* leaves are shown in Fig. 2A. The processed spectra, SG, SNV, MSC, 1-Der, SNV+TB, SG+1-Der, and MSC+1-Der values are shown in Figs. 2B–2H, respectively. The raw spectra appeared homogeneous, as can be seen by visual inspection of the data in Fig. 2A. As shown in Fig. 2A, the spectral curves are relatively flat between 450 and 500 nm, but the raw spectra between 500 and 600 nm show significantly different features and a notable decrease in the green range around 550 nm with increased anthocyanin content.

### Statistical values of properties of interest

The statistics of the measured anthocyanin content for the 114 *P. cerasifera* leaf samples determined in this study are listed in Table 1 and include the minimum, maximum, mean, standard deviation (S.D.), and number of samples for the different data sets. The reference values of anthocyanin content exhibited a broad range of variation, a result that facilitated calibration.

**Table 2 Prediction results of anthocyanin content by PCR with different preprocessing in calibration and validation sets.**

| Pretreatment | PCs | Calibration | | | Validation | | |
|---|---|---|---|---|---|---|---|
| | | $R^2_{cal}$ | $RMSE_c$ | $RPD_{cal}$ | $R^2_{val}$ | $RMSE_p$ | $RPD_{val}$ |
| Raw | 5 | 0.777 | 0.462 | 2.117 | 0.743 | 0.477 | 1.973 |
| SNV | 5 | 0.934 | 0.250 | 3.911 | 0.888 | 0.315 | 2.988 |
| MSC | 7 | 0.915 | 0.286 | 3.419 | 0.844 | 0.372 | 2.530 |
| SG | 5 | 0.776 | 0.463 | 2.112 | 0.741 | 0.479 | 1.965 |
| 1-Der | 6 | 0.810 | 0.427 | 2.290 | 0.843 | 0.373 | 2.523 |
| MSC+1-Der | 8 | 0.881 | 0.337 | 2.902 | 0.881 | 0.337 | 2.793 |
| SNV+BS | 5 | 0.933 | 0.253 | 3.865 | 0.864 | 0.347 | 2.712 |
| SG+1 −Der | 8 | 0.857 | 0.370 | 2.643 | 0.864 | 0.348 | 2.705 |

## PCR models

PCR analysis was applied for the calibration and prediction of anthocyanin content. Eight different models for anthocyanin content were developed with different spectra. Different PCs were applied to build the optimal calibration models. The prediction results of the calibration and validation sets are shown in Table 2. Comparison of these models show that the spectra preprocessed by SNV displayed the best performance for anthocyanin content prediction. The values of $R^2_{val}$, $RMSE_p$, and $RPD_{val}$ in the validation set from the optimal PCR model were 0.888, 0.315, and 2.988, respectively. This prediction accuracy was therefore classified as very good. The performances using SG and Raw were poor, with the $R^2_{val}$ and $RPD_{val}$ for both models that were lower than 0.80 and 2.0, respectively. According to the aforementioned criteria, we can only say that these two models might be of some value in quantitative prediction of anthocyanin content. However, the other five PCR models yielded $RPD_{val}$ values above 2.5 and the $R^2_{val}$ values in the range of $0.80 \leq R^2 < 0.90$, which indicated the suitability of these models for very good quantitative predictions of leaf anthocyanin content. Figure 3A shows the reference versus predicted value plots for anthocyanin content using the optimal PCR model. The closer the distance the sample points are to this solid line represents better predictive results. As indicated in Fig. 3A, the sample points in the calibration and validation sets were distributed near, but not tightly close to the ideal line. Also, several dots were lovated far from the ideal line, indicating a large predictive error.

## PLSR models

Partial least squares regression (PLSR) models using the pretreatment spectra were also tested and the results are shown in Table 3. According to the results, the optimal preprocessing for anthocyanin content also was SNV, based on the values of the prediction performance evaluation indices. The values of the optimal determination coefficients $R^2_{val}$, $RMSE_p$, and $RPD_{val}$ for the validation set were respectively 0.901, 0.295 and 3.191. This prediction accuracy was classified as excellent. The performance using MSC+1-Der was the worst of the tested models, with the smallest predicted $R^2_{val}$ and $RPD_{val}$ values and the largest $RMSE_p$ values. Overall, the $RPD_{val}$ values above 2.0 and the $R^2_{val}$ values above 0.8 for all PLSR models indicated that these models provide very good quantitative

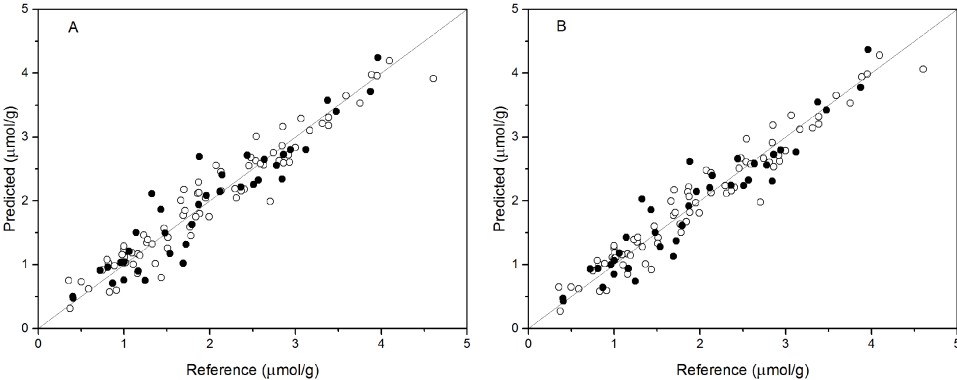

**Figure 3 Measured vs. predicted values for anthocyanin content obtained by the best PCR model (A) and PLSR model (B).** Black open circles represent calibration samples and solid circles represent validation samples. The solid lines correspond to the ideal results which meant the predicted values were equal to the reference values.

**Table 3 Prediction results of anthocyanin content by PLSR with different preprocessing in calibration and validation sets.**

| Pretreatment | LVs | Calibration | | | Validation | | |
|---|---|---|---|---|---|---|---|
| | | $R^2_{cal}$ | $RMSE_c$ | $RPD_{cal}$ | $R^2_{val}$ | $RMSE_p$ | $RPD_{val}$ |
| Raw | 9 | 0.933 | 0.254 | 3.850 | 0.873 | 0.336 | 2.801 |
| SNV | 5 | 0.943 | 0.233 | 4.197 | 0.901 | 0.295 | 3.191 |
| MSC | 4 | 0.894 | 0.318 | 3.075 | 0.847 | 0.368 | 2.558 |
| SG | 9 | 0.928 | 0.262 | 3.732 | 0.878 | 0.329 | 2.861 |
| 1-Der | 5 | 0.886 | 0.330 | 2.963 | 0.882 | 0.323 | 2.914 |
| MSC+1-Der | 5 | 0.921 | 0.274 | 3.569 | 0.802 | 0.419 | 2.246 |
| SNV+BS | 5 | 0.943 | 0.234 | 4.179 | 0.891 | 0.311 | 3.026 |
| SG+ 1-Der | 5 | 0.884 | 0.332 | 2.945 | 0.883 | 0.323 | 2.914 |

predictions for leaf anthocyanin content. The plot of reference versus predicted values for anthocyanin content using the optimal PLSR model is shown in Fig. 3B. The sample points in the calibration and validation sets are distributed much closer to the ideal line, but there was still a large deviation between the predicted values and the actual value in the PLSR models. Although according to the evaluation criteria, the optimal PLSR model should be an excellent model/predictor, the results showed that it was not ideal for use in practical analysis.

## BPNN models

The performance of BPNN models was next validated using the validation set, and the prediction results are shown in Table 4 and Fig. 4. As shown in Table 4, the values of $R^2_{val}$, $RMSE_p$, and $RPD_{val}$ in the validation set were 0.922, 0.270, and 3.489, respectively, for the BPNN-LVs model and 0.920, 0.274, and 3.439, respectively, for the BPNN-PCs model. Based on these values, both models showed excellent prediction accuracy. Very small differences in $R^2$, $RMSE_p$ and RPD values were observed between the BPNN-LVs

**Table 4  Prediction results of anthocyanin content by BPNN models in calibration and validation sets.**

| Model | Calibration | | | Validation | | |
|---|---|---|---|---|---|---|
| | $R^2_{cal}$ | $RMSE_c$ | $RPD_{cal}$ | $R^2_{val}$ | $RMSE_p$ | $RPD_{val}$ |
| BPNN-PCs | 0.958 | 0.203 | 4.648 | 0.920 | 0.274 | 3.439 |
| BPNN-LVs | 0.961 | 0.195 | 4.819 | 0.922 | 0.270 | 3.489 |

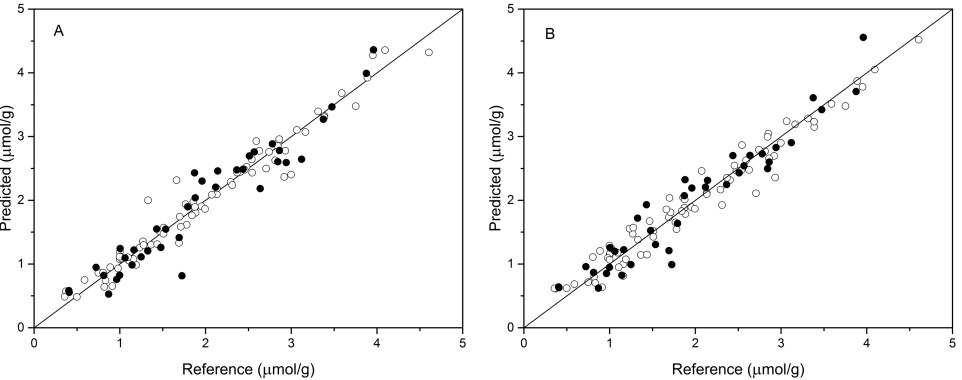

**Figure 4  Measured vs. predicted values for anthocyanin content obtained by BPNN-PCs model (A) and BPNN-LVs model (B).** Black open circles represent calibration samples and solid circles represent validation samples. The solid lines correspond to the ideal results which meant the predicted values were equal to the reference values.

model and the BPNN-PCs model. The performance of the BPNN-LVs model was a little better than that of the BPNN-PCs model. The plots of reference versus predicted values points for anthocyanin content using the BPNN models are shown in Fig. 4. The sample plots were tighter about the ideal line than those obtained using the PCR and PLSR models (see Fig. 3). The results show that the BPNN models outperformed the PCR and PLSR models, with very good agreement between the predicted values and the actual values in the BPNN models. This high prediction precision could satisfy the accuracy standards for practical applications and these results should support further research of in-field detection methods for anthocyanin content in plant leaves.

## DISCUSSION

The raw spectra of *P. cerasifera* leaves between 500 and 600 nm show a notable decrease in the green range around 550 nm with increase of anthocyanin content. The main spectral feature of anthocyanin absorption in vivo is a peak around 550 nm; consistent with the finding of *Gitelson, Merzlyak & Chivkunova (2001)* that the peak magnitude was closely related to anthocyanin content. In this study, three calibration methods were tested using all of the spectral reflectance of the selected wavebands to build models. The selected wavebands should be sensitive to the anthocyanin, and insensitive to chlorophyll, water, and the effects of leaf structure, and the wavebands between 450 and 600 nm meet this requirement. The study results showed that spectral reflectance between 450 and 600 nm

well-predicted leaf anthocyanin content in *P. cerasifera*. Other studies have also used the visible wavelength bands to predict leaf anthocyanin content (e.g., *Gitelson, Merzlyak & Chivkunova, 2001*; *Gitelson et al., 2006*; *Steele et al., 2009*; *Garriga et al., 2014*).

In addition, as shown in Tables 2 and 3, comparison of the results using the same pretreatments in the PCR and PLSR models, the difference values of $R^2$, RMSE, and RPD were greater than 0.05, 0.06, and 0.8, respectively, for the calibration set and predicted values of most models. The better results for the calibration set indicate that the calibration model was not very stable. The sample points for the calibration and validation sets of the PLSR model are distributed much closer to the ideal line than those of the PCR model (Figs. 3A and 3B), indicating that the PLSR model outperformed the PCR model. Comparison of the prediction results of PCR and PLSR models with the same pretreatment reveals better performance of PLSR models compared to that of the PCR models, which is consistent with the results of another study (*Vasques, Grunwald & Sickman, 2008*). This may be because the PLSR model can simultaneously consider the spectral data matrix (X) and the target chemical properties matrix (Y) (*Liu & Liu, 2013*). Of the BPNN models, the performance of the BPNN-LVs model was a little better than that of the BPNN-PCs model. *Mouazen et al. (2010)* reported similar results for the prediction of selected soil properties using Vis/NIR spectroscopy.

Both the leave-one-out cross-validation and predictive results showed that the BPNN model outperformed the PCR and PLSR models (Tables 2–4, and Figs. 3 and 4). The result is consistent with results from other studies of VNIRS of predictions for total anthocyanin content in new-season red-grape homogenates with PLSR and ANN (*Janik et al., 2007*). Additionally, *Liu et al. (2008)* reported similar results for the determination of acetolactate synthase activity and protein content of oilseed rape (Brassica napus L.) leaves using Vis/NIR spectroscopy. *Janik, Forrester & Rawson (2009)* and *Mouazen et al. (2010)* also reported similar results for the prediction of selected soil chemical and physical properties using mid-infrared or Vis/NIR spectroscopy. The higher performance of the BPNN model may be because it can the nonlinear relationship typical of spectrum analysis, while PLSR and PCR models, which are built upon a linear algorithm, do not consider certain latent nonlinear information in the spectral data (*Li & He, 2010*). The performance of the BPNN-LVs model was a little better than that of the BPNN-PCs model according to the $R^2$, $RMSE_p$, and RPD values. *Mouazen et al. (2010)* reported similar results for the prediction of selected soil properties using Vis/NIR spectroscopy. Thus, we have demonstrated the feasibility of using spectral reflectance between 450 and 600 nm to estimate leaf anthocyanin content in *P. cerasifera* under laboratory conditions. Of cause, the canopy architecture of plants may be very complex under field conditions. In future work, additional samples and samples of different species samples should be prepared for calibration based on both laboratory and field conditions to expand testing of the BPNN-LVs model and improve model stability for future practical applications. Additionally, chlorophyll's interference should be considered for samples with low to moderate anthocyanin content (*Gitelson, Chivkunova & Merzlyak, 2009*). Future work could be done to discover useful information or effective wavelengths or wavebands for the non-destructive determination of anthocyanin content of plants.

## CONCLUSIONS

The anthocyanin content was successfully determined by spectral reflectance between 450 and 600 nm combined with chemometric methods. In the PCR and PLS models, spectra the preprocessed by SNV achieved the best performance for the prediction of anthocyanin content. Acceptable prediction accuracies were achieved by the PCR and PLS models, but this level of accuracy may be not satisfactory for practical applications. The performance of the PLSR models was better than that of the PCR models, but the BPNN models showed greatly improved predictive capacity. The two BPNN models were developed for the prediction of anthocyanin content outperformed the PCR and PLSR models. The $R^2_{val}$, $RMSE_p$, and $RPD_{val}$ values for the validation set using the BPNN-LVs model were 0.922, 0.270, and 3.489, respectively, and those of the BPNN-PCs model were 0.920, 0.274, and 3.439, respectively. Thus, the performance of the BPNN-LVs model was best. The results indicate that visible spectroscopy combined with BPNN calibrations can successfully determine the leaf anthocyanin content in *P. cerasifera*. Based on the results achieved in this study, it is recommended to adopt BPNN-LVs analysis as the best modeling method to predict plant leaf anthocyanin content. The use of spectral reflectance data between 450 and 600 nm here represents a significant contribution to methods for the nondestructive determination of leaf total anthocyanin content.

## ACKNOWLEDGEMENTS

We thank Xiaoxing Wang and Li Wang of Northwest A&F University for assistance in examining and measuring specimens.

### Funding

This work was supported by the Henan Science and Technology Plan Program (182102110206), the Doctoral Scientific Research Foundation of Henan University of Science and Technology of China (13480074). The funders had no role in study design, data collection and analysis, decision to publish, or preparation of the manuscript.

### Grant Disclosures

The following grant information was disclosed by the authors:
Henan Science and Technology Plan Program: 182102110206.
Doctoral Scientific Research Foundation of Henan University of Science and Technology of China: 13480074.

### Competing Interests

The authors declare there are no competing interests.

### Author Contributions

- Xiuying Liu conceived and designed the experiments, performed the experiments, analyzed the data, contributed reagents/materials/analysis tools, prepared figures and/or tables, authored or reviewed drafts of the paper, approved the final draft.

- Chenzhou Liu performed the experiments, contributed reagents/materials/analysis tools, authored or reviewed drafts of the paper, approved the final draft.
- Zhaoyong Shi performed the experiments, analyzed the data, contributed reagents/materials/analysis tools, prepared figures and/or tables, authored or reviewed drafts of the paper, approved the final draft.
- Qingrui Chang conceived and designed the experiments, authored or reviewed drafts of the paper, approved the final draft.

## Data Availability

The raw measurements are available in the Supplemental Files.

## Supplemental Information

Supplemental information for this article can be found online at http://dx.doi.org/10.7717/peerj.7997#supplemental-information.

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
