# Peer review of "Comparison of prediction power of three multivariate calibrations for estimation of leaf anthocyanin content with visible spectroscopy in Prunus cerasifera"

_PeerJ, doi:10.7717/peerj.7997_

## Round 0.1 · original submission · Major Revisions

· Academic Editor

Major Revisions

Please respond in detail to the reviewer comments especially those of Reviewer 2, regarding models validation

Reviewer 1 ·

Basic reporting

The manuscript is well written with sufficient references and field background. In addition, sufficient data (figures and tables) were also provided.

Experimental design

- Research question well defined.
- Rigorous investigation performed to high technical and ethical standard
- However, method description should be improved (refer to my comments).

Validity of the findings

The findings are very useful for future research, however, supporting data should be provided (please refer to my comments).

Additional comments

Reviewer’s general comments: The manuscript reports on the application of multivariate data analysis to predict anthocyanin content in the leaf of Prunus cerasifera. It is quite interesting paper comparing three methods: PCR, PLSR, and BPNN modelling for their prediction capacity. However, there are some points need to be addressed

Line 114: How many gram is the leaf samples? Please mention
Line 115: Please add time of extraction (soaking)
Line 134: Where are the results for correlation analysis? I don’t find in the table or figure
Line 156: What is the software used to construct the PCR and PLSR models? Should be mentioned
Line 175-176: How to assign the samples as calibration set and validation set? Selection method?
Line 216: “RPDval values above 2.5 and the R2val values below 0.9 for the other 5 PCR models indicated that very good quantitative predictions”
Shouldn’t the higher value of R2 is the better? I suggest to change this statement to
“RPDval values above 2.5 and the R2val values in range of 0.80≤R2<0.90 for the other 5 PCR models indicated that very good quantitative predictions”
Line 279-281: Which results saying calibration and predicted of PCR and PLSR models were significantly different? Not clear. And then the comparison is between the same pretreatment in the same model or between the different pretreatments in the same model? Its better to show the sig value (pvalue) of this comparison data.

·

Basic reporting

The manuscript describes a comparative study for evaluating three different chemometric models used for quantitative determination of anthocyanin content in Prunus cerasifera leaves based on their UV Vis measurements. Moreover, different preprocessing methods were also evaluated to determine the optimal spectral pretreatments. The manuscript is neat and well written however some points are needed to be considered:
In the introduction section, the authors should explain the importance of using the multivariate models and their potential to determination different chemical classes in plants and with special focus on coupling of UV Vis spectrophotometry with the multivariate analysis as this is the core of the manuscript.

Experimental design

no comment

Validity of the findings

My major concern is regarding non-linear neural network model that has been applied besides PCR/PLS models. Either the studied data is linear, and hence PCR/PLS should be the calibration tool of choices, or the data is not linear, in which case it might be justified to move to a non-linear calibration model. The lack of linearity could be checked in the multivariate case with suitable statistical tests [ Anal. Chim. Acta 552 (2005) 25-35; Anal. Chim. Acta 376 (1998) 153-168]. The results of these tests, combined with a statistically significant improvement in prediction error by applying non-linear models, might constitute a proof that the system behaves in a non-linear manner. One particularly appealing technique is based on the augmented partial residual plots (APARP) which can be constructed for PLS models [Detection of nonlinearity in multivariate calibration, Anal. Chim. Acta 376 (1998) 153-168]. The following is the recommended protocol: (1) model the multivariate data with A latent variables, (2) regress analyte concentrations against an augmented model including the A scores and the squared values of the first score, (3) plot the portion of the concentration data modelled by the first score and the squared first score vs. the first score, (4) compute the residuals of the linear regression of the latter plot, and finally (5) check for the presence of correlations in the residuals of the latter regression using a suitable statistical test. If significant trends are found in these APARP residuals, the data can be considered non-linear.
Another important issue is the poor optimization of the neural networks where the authors used just 10 nodes between the first and second layer based on a previous study on soil properties using visible and near infrared spectroscopy which is not acceptable. The machine learning algorithms usually suffer from over fitting which required careful optimization of the parameters which in case of ANN are number of nodes, learning rates, momentum, transfer function and the training algorithm.
Finally, the performance of these models should be carefully checked through suitable statistical procedures. It is necessary to check whether the RMSE for test samples is indeed smaller (statistically significantly or not).

Reviewer 3 ·

Basic reporting

There are some typos and formatting that need amendment. We advise the author to care for tense throughout the manuscript. Also, references need to be revised for formatting and uniformity.

Experimental design

No comment

Validity of the findings

We would like to see more supporting literature on the idea of the and impact of the work. Are there valid problems for anthocyanin assessment? What value does the paper add? Does it save time? Solvents? Is it the software available for use? Are the results repeatable?

Additional comments

The work as presented deals with one species and optimizes or encourages the use of a certain chemometric model and preprocessing for predicting anthocyanin content spectrophotmetrically. But how can we make sure that this model is rigorous and repeatable if we used other species? If there were economic importance for the studied species, then I would understand the importance of the work. But since this is just a study of optimizing anthocyanin content using one example, emphasis has to be made on the importance of applying this method on other anthocyanin containing plants.
Also, why did the author select these 3 chemometric models for comparison?

---

## Round 0.2 · accepted · Accept

· Academic Editor

Accept

The authors have adequately addressed reviewer comments in the revised version